# Efficacy and safety of Galgeun-tang-ga-cheongung-sinyi for nasal congestion with common cold: A study protocol for randomized, double-blind, placebo-controlled, parallel, multicenter clinical trial

**Seong-Cheon Woo**[1☯‡], **Yee Ran Lyu**[2☯‡], **Byong-Kab Kang**[2], **Bok-Nam Seo**[3], **Ae-Ran Kim**[3], **Boram Lee**[2], **Yang Chun Park**[1*]

1 Division of Respiratory Medicine, Department of Internal Medicine, College of Korean Medicine, Daejeon University, Daejeon, Republic of Korea, 2 KM Science Research Division, Korea Institute of Oriental Medicine, Daejeon, Republic of Korea, 3 Clinical Medicine Division, R&D Strategy Division, Korea Institute of Oriental Medicine, Daejeon, Republic of Korea

☯ These authors contributed equally to this work.
‡ These authors share first authorship on this work.
* omdpyc@dju.kr (YCP)

## Abstract

### Background

Common cold is a common respiratory disease caused by a viral infection of the upper respiratory tract, leading to various symptoms including cough, sneezing, nasal congestion, and sore throat. As the medications used to treat the common cold may cause adverse effects such as gastrointestinal symptoms, drowsiness, and decreased concentration, it is necessary to explore alternative treatments for patients with the common cold. The aim of this study is to assess the effectiveness and safety of Galgeun-tang-ga-cheongung-sinyi (GGTCS), an herbal formula used to treat respiratory diseases, as an alternative medicine for nasal congestion with common cold.

### Methods

This is a randomized, double-blind, placebo-controlled, parallel, multicenter clinical trial. A total of 176 participants will be recruited and randomly allocated to the GGTCS group and control (placebo) group in a 1:1 allocation ratio. GGTCS or placebo will be administered three times per day for 7 days. The participants will be instructed to discontinue the medicine if the symptoms of the common cold disappear within 7 days of the treatment period and return the remaining medicine to the investigators. The primary outcome measures are changes in the total score of the Wisconsin Upper Respiratory Symptom Survey-21-Korean version (WURSS-21-K) (symptom score + quality of life score) on day 7 compared with baseline. Secondary outcome

**Data availability statement:** No datasets were generated or analysed during the current study. All relevant data from this study will be made available upon study completion.

**Funding:** This research was supported by a grant from the Korea Health Technology R&D Project through the Korea Health Industry Development Institute (KHIDI), funded by the Ministry of Health & Welfare, Republic of Korea (RS-2024-00441852). The funders had no role in study design, data collection and analysis, decision to publish, or preparation of the manuscript.

**Competing interests:** The authors declare no competing interests.

measures include the total, symptom, and quality of life scores of the WURSS-21-K, nasal congestion severity score, nasal and systemic symptom score visual analogue scale (VAS), duration of common cold symptoms, recovery from common cold, global evaluation of efficacy, and 5-level EuroQol 5-dimensional questionnaire (EQ-5D-5L). The exploratory outcome measure is a questionnaire for common cold pattern identification (QCCPI), and the safety assessment factors include adverse effects, vital signs, electrocardiograms, and clinical laboratory tests.

## Discussion

The goal of this study is to assess the effectiveness and safety of GGTCS for nasal congestion with the common cold and the clinical use of herbal medicine. The results are expected to help researchers design further clinical trials using herbal medicines to treat the common cold.

## Trial registration

This trial was registered on February 28, 2025 at the National Clinical Trial Registry Clinical Research Information Service (https://cris.nih.go.kr) with the identifier number KCT0010251.

## Introduction

The common cold, an illness caused by acute viral infection of the upper respiratory tract including nose, sinuses, pharynx, and larynx, is a self-limiting disease that can be cured spontaneously without treatment [1]. Although the symptoms of the common cold are mild in most cases, it causes suffering in individuals and is an economic burden on societies owing to its high prevalence [2]. In the United States, the common cold is the most common illness, with 25 million cases per year [3] and the average loss due to absenteeism and productivity reduction is estimated to be 8.7 working hours with every episode of the common cold yielding an economic loss of approximately $25 billion per year [4]. In South Korea, approximately 4.5 million outpatients were treated in 2023 for the common cold [5].

Common cold symptoms include rhinorrhea, nasal congestion, cough, sneezing, sore throat, muscle aches, and fever [6]. These symptoms can lead to psychological effects, such as general discomfort, mood changes, and attention deficits, which reduce the quality of life and ability to work [6,7]. Moreover, the common cold can cause complications such as secondary bacterial infections and exacerbations of asthma and chronic obstructive pulmonary disease (COPD), which may be severe in patients with respiratory diseases [1,8,9]. Analgesics, decongestants, antihistamines, antitussives, and expectorants are commonly used to treat the common cold [10]. However, nonsteroidal anti-inflammatory drugs (NSAIDs) can cause gastrointestinal symptoms including dyspepsia and gastric ulcers [11], and first-generation antihistamines have side effects such as fatigue, drowsiness, and decreased concentration

[12]. Decongestants can cause nausea, vomiting, insomnia, and urinary dysfunction, whereas long-term use of nasal decongestants can cause rebound nasal congestion [13]. Intranasal corticosteroids are associated with local adverse effects, including throat irritation, nasal dryness, and epistaxis [14]. Although nasal irrigation is widely used intervention for upper respiratory tract infections, it has risks of nasal discomfort, otalgia, and microbial contamination from solutions or devices [15]. Therefore, it is necessary to explore safe alternative treatments that are effective against nasal congestion in common cold.

In previous clinical trials, it has been shown that herbal medicines are clinically effective in treating the symptoms of the common cold [16–18]; however, the clinical effects on specific symptoms have not been evaluated. Accordingly, the focus of the present study is the effects of an herbal treatment on nasal congestion, given that this symptom can lead to complications such as sinusitis and otitis media, as well as contributing to sleep disturbance, fatigue, poor concentration, and decreased productivity [19,20].

In Korean medicine, additional medicinal herbs are added to existing herbal formula depending on the symptoms. Galgen-tang (GGT), also known as Gegen-Tang in Chinese and Kakkon-to in Japanese, is an herbal formula with anti-inflammatory and antipyretic effects and has been used to treat viral infectious diseases such as common cold, influenza, and measles [21]. Galgeun-tang-ga-cheongung-sinyi (GGTCS), known as Gegen-Tang-Jia-Chuan-Xiong-Xin-Yi in Chinese and Kakkon-to-ka-senkyu-shin'i in Japanese, is an herbal formula composed of two medicinal herbs (*Cnidii Rhizoma* and *Magnoliae Flos*) added to GGT and is used for the treatment of nasal congestion related to the common cold and rhinitis [22,23]. GGT is used to improve the symptoms of the common cold, and GGTCS, a modified GGT with the addition of two medicinal herbs, is used to alleviate nasal congestion associated with the common cold. In previous studies, it was reported that GGTCS has an immediate effect on nasal obstruction induced by allergy [24] and a clinical trial has been designed to investigate the safety and efficacy of GGTCS for patients with chronic rhinosinusitis [25]. Therefore, we hypothesized that GGTCS has the potential to treat the symptoms of common cold, especially nasal congestion. However, previous studies have primarily focused on allergic rhinitis and chronic rhinosinusitis, and evidence supporting the use of GGTCS for the treatment of the common cold remains limited. The available evidence is largely restricted to case reports and clinical trial protocol [24–26]. Therefore, we aimed to investigate the efficacy and safety of GGTCS for nasal congestion associated with the common cold by assessing the severity of symptoms, duration of the common cold, and quality of life through a randomized controlled trial.

## Methods

### Study design

This is a randomized, double-blind, placebo-controlled, parallel, multicenter clinical trial, with the aim of assessing the efficacy and safety of GGTCS for nasal congestion associated with the common cold. The trial will be conducted at three sites: Daejeon University Korean Medicine Hospital, Kyung Hee University Korean Medicine Hospital, and Pusan University Korean Medicine Hospital. In this trial, 176 participants will be recruited, screened and randomly assigned to the GGTCS and placebo groups in a 1:1 allocation ratio. Each group will be given the investigational medicines and take them three times a day as long as the symptoms persist up to a maximum of 7 days. The detailed study design is summarized in Fig 1 and 2.

**Inclusion criteria.**

1) Men and women aged between 19 and 75 years;

2) Diagnosed with the common cold, symptoms occurring within 72 hours of the start of the trial;

3) More than two symptoms related to the common cold including runny nose, nasal congestion, sneezing, sore throat, cough, headache, chills, and body aches;

| STUDY PERIOD | | | |
|---|---|---|---|
| | **Enrolment** | **Allocation** | **Post-allocation** |
| **VISIT** | Visit 1 (Screening[1]) | Visit 2 | Visit 3 |
| **TIMEPOINT** | Day -1-0 | Day 0 | Day 7 + 3 days |
| **ENROLLMENT** | | | |
| Informed consent | X | | |
| Eligibility screen | X | | |
| Demographic characteristics | X | | |
| Medical history | X | | |
| Physical examination | X | | |
| Electrocardiogram | X | | |
| Chest X-ray | △ | | |
| Paranasal Sinuses X-ray | △ | | |
| Vital signs | X | X | X |
| Allocation | | X | |
| **INTERVENTIONS** | | | |
| GGTCS | | ←——————————————→ | |
| Placebo | | ←——————————————→ | |
| **ASSESSMENTS** | | | |
| Clinical laboratory test[2] | X | | X |
| Pregnancy diagnosis test | △ | | |
| Nasal congestion severity score | X | Daily | |
| | | X | X |
| WURSS-21-K[3] | | Daily | |
| | | X | X |
| Nasal symptom score VAS[3] | | Daily | |
| | | X | X |
| Systemic symptom score VAS[3] | | Daily | |
| | | X | X |
| Duration of common cold symptoms[3] | | Daily | |
| | | | X |
| Recovery of common cold | | | X |
| Global evaluation of efficacy | | | X |
| EQ-5D-5L | | X | X |
| PICCQ | X | | |
| Adverse reaction assessment | | X | X |
| Compliance test | | X | X |

**Fig 1. Schedule of enrollment, interventions and outcome measurements.** X: All participants; △: If considered necessary by the investigator. 1 Screening test should be conducted within one day before Visit 2 and the results of clinical laboratory test should be able to be checked at Visit 2. 2

CBC (erythrocyte count, leukocyte count, hemoglobin, and hematocrit); LFT (AST, ALT, BUN, creatinine, and glucose). 3 These will be assessed through evaluation diary: 7-day self-recording worksheets will be provided at baseline after registration. The participants will record data daily and return them to the investigators at Visit 3. ALT, alanine aminotransferase; AST, aspartate aminotransferase; BUN, blood urea nitrogen; CBC, complete blood count; EQ-5D-5L, 5-level EuroQol 5-dimensional questionnaire; GGTCS, Galgeun-tang-ga-cheongung-sinyi; LFT, liver function test; PICCQ, pattern identification for chronic cough questionnaire; VAS, visual analog scale; WURSS-21-K, Wisconsin upper respiratory symptom survey-21-Korean version.

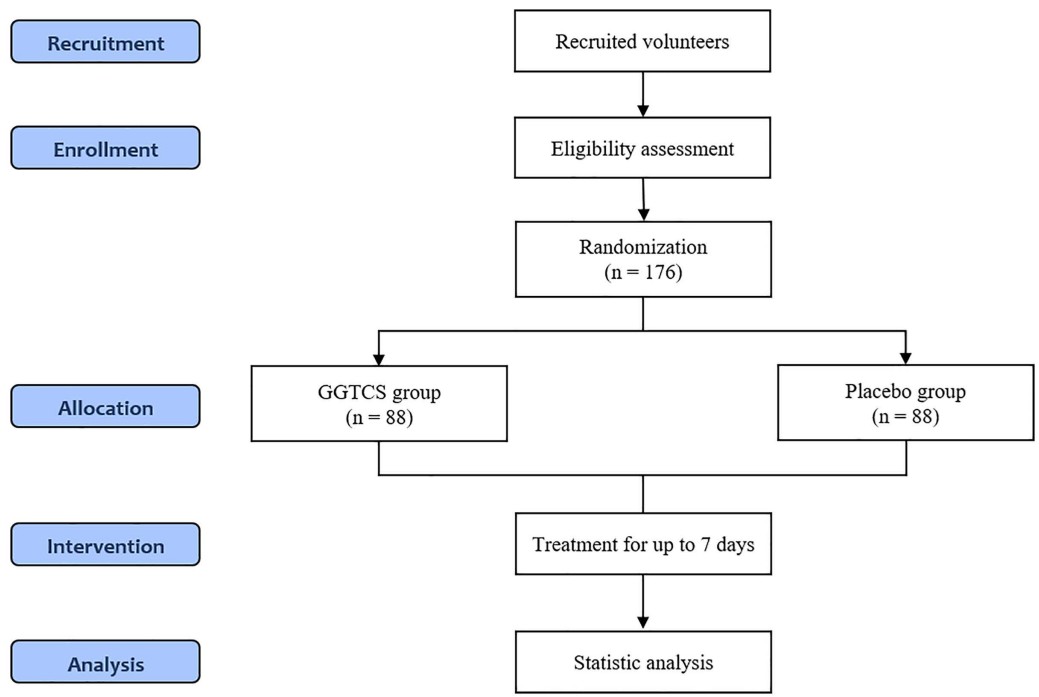

**Fig 2. Flowchart of the study plan.** GGTCS, Galgeun-tang-ga-cheongung-sinyi.

4) Nasal congestion severity score greater than 2;

5) Those who voluntarily decide to participate in this clinical trial and give written informed consent to abide by the precautions after listening a detailed explanation of this clinical trial and fully understanding it.

   **Exclusion criteria.**

1) Those with sinusitis, allergic rhinitis, pneumonia, influenza, COVID-19, cough or sore throat with sudden fever above 38 °C, bronchitis, otitis media, tonsillitis (if accurate examination is necessary, paranasal sinuses (PNS) view or chest X-ray will be conducted).

2) Diagnosis of chronic respiratory disease (chronic obstructive pulmonary disease, bronchial asthma, bronchiectasis, interstitial lung disease, and other chronic respiratory diseases);

3) Those who had taken antibiotics, antivirals, steroids, nasal decongestants, antihistamines, antitussives, expectorants, or other medications to alleviate symptoms of the common cold within 1 week of screening;

4) Liver or renal impairment (alanine aminotransferase, aspartate aminotransferase, and creatinine ≥ 3 times the upper normal limit at screening);

5)  Those with generic disorders such as galactose intolerance, Lapp lactase deficiency, or glucose-galactose malabsorption;

6)  Pseudo-aldosteronism or myopathy induced by hypokalemia;

7)  Comorbidities that interrupt the treatment of cancers or clinically significant disorders of the kidney, liver, psychiatric system, cardiovascular system, respiratory system, endocrine system, or central nervous system thus interfering with the assessment of efficacy and safety of the investigational drug or completion of the clinical study;

8)  Unregulated hypertension (systolic blood pressure ≥ 160 mmHg or diastolic blood pressure ≥100 mmHg);

9)  Unregulated diabetes mellitus (fasting blood sugar ≥ 180 mg/dL);

10) History of hypersensitivity reaction to materials of drugs used in this clinical trial;

11) History of alcoholism or substance abuse;

12) Current smokers or those with history of smoking more than 30 packs/year;

13) Pregnant or lactating women;

14) Those who do not use medically acceptable contraception (e.g., intrauterine device with proven pregnancy failure rate in spouse or partner, simultaneous use of barrier method for men or women with spermicide, or surgical procedures, such as vasectomy, tubectomy, tubal ligation, or hysterectomy for oneself or one's partner) during the clinical trial;

15) Those who participated in other clinical trials within 30 days before participation in this clinical trial;

16) Those determined by investigators to be ineligible to participate in this trial.

## Sample size

The objective of this clinical trial is to evaluate the efficacy and safety of GGTCS for the nasal congestion with common cold compared with a placebo. The primary outcome of this clinical trial is the between-group difference in total WURSS-21-K (Wisconsin Upper Respiratory Symptom Survey-21-Korean version) score measured at day 7, adjusted for baseline values. The sample size was determined based on a previous clinical trial that evaluated the efficacy of herbal medicine for the common cold using WURSS-21-K [17]. With reference to the results of a previous study, the mean difference was set to 10 and the standard deviation was set to 21. Adopting a significance level of 5% (α) and a power of 80% (1−β), the calculated sample size was 70 participants in each of the two groups (GGTCS and placebo group) with a 1:1 allocation ratio.

Hypothesis:

$H_0$: $\mu_t = \mu_c$ vs. $H_1$: $\mu_t \neq \mu_c$

The sample size in each group was calculated as follows:

$$\left\{ \frac{2\left(z_{1-\frac{\alpha}{2}} + z_\beta\right)^2 * \sigma^2}{\left|\mu_t - \mu_c\right|} \right\} = \left\{ \frac{2(1.96 + 0.84)^2 * 21^2}{10^2} \right\} = 69.227 \approx 70$$

Considering a dropout rate of 20%, 176 participants were required (88 per group).

* $\mu_t$: Mean difference in WRUSS-21-K score at day 7 compared with baseline in GGTCS group
* $\mu_c$: Mean difference in WRUSS-21-K score at day 7 compared with baseline in control (placebo) group

The participants will be recruited from the following three sites: the Daejeon University Korean Medicine Hospital, the Kyung Hee University Korean Medicine Hospital, the Pusan University Korean Medicine Hospital by competitive registration.

## Recruitment

Participants will be recruited from regular promotions through mass media, such as newspaper leaflets, daily newspapers, advertising posters, and brochures displayed in the hospital's outpatient departments. If the recruitment of participants is insufficient during the clinical trial, additional advertising will be carried out in the local community; for example, advertisements in public transportation and apartment buildings.

## Interventions

Participants will be instructed to take the investigational medicine (GGTCS or placebo) before or between meals, three times per day for up to 7 days. In this trial, 21 packs of the investigational medicine will be packaged as 7-day doses. Participants will be instructed that if the symptoms of the common cold resolve before 7 days, they should stop taking the investigational medicine and return the remainder to the investigators. To confirm compliance, the investigators will check whether participants have taken the investigational medicine and the remainder at visit 3. The investigational medicine (GGTCS) consists of nine medicinal herbs: *Puerariae Radix* 2.67 g, *Ephedrae Herba* 1.33 g, *Cinnamomi Ramulus* 1.00 g, *Paeoniae Radix* 1.00 g, *Glycyrrhizae Radix et Rhizoma* 0.67 g, *Zingiberis Rhizoma* 0.33 g, *Zizyphi Fructus* 1.33 g, *Cnidii Rhizoma* 1.00 g, and *Magnoliae Flos* 1.00 g, which is manufactured as light brown granules. GGTCS will be purchased from the Kyungbang Pharm Corporation (Incheon, Republic of Korea), a company that has obtained authorization from the Korea Good Manufacturing Practice. The placebo will be manufactured by National Institute for Korean Medicine Development as light brown granules. The ingredients and compositions of the GGTCS and placebo are presented in Table 1.

## Outcome measures

**Primary outcome measures.** The primary outcome in this study is the change in the total WURSS-21-K score (symptom score + quality of life score) on day 7 compared with the baseline. The WURSS-21 has been used to assess the severity of common cold symptoms and quality of life, and its reliability and validity were proved [27]. The WURSS-21-K is the Korean version of the WURSS-21, the reliability and validity of which have been established [28]. The WURSS-21-K consists of 21 questions: ten questions on the symptoms of the common cold, nine on functional quality of life, one on the overall severity of the common cold, and one on overall changes in condition. The questions, except for one question on overall changes in the condition, will be recorded on a seven-point Likert scale (0 = none; 1 = very mild; 3 = mild; 5 = moderate; 7 = severe). In the question on overall changes in condition, participants will report their changes in condition using seven options: very much better, somewhat better, a little better, the same, a little worse, somewhat worse, and very much worse. Because the question on overall changes in condition is irrelevant to the remaining 20 questions, we will

**Table 1. Composition of Galgeun-tang-ga-cheongung-sinyi (GGTCS).**

| Latin name | Amount (g) |
|---|---|
| **Galgeun-tang-ga-cheongung-sinyi (GGTCS)** | |
| *Puerariae Radix* | 2.67 |
| *Ephedrae Herba* | 1.33 |
| *Cinnamomi Ramulus* | 1.00 |
| *Paeoniae Radix* | 1.00 |
| *Glycyrrhizae Radix et Rhizoma* | 0.67 |
| *Zingiberis Rhizoma* | 0.33 |
| *Zizyphi Fructus* | 1.33 |
| *Cnidii Rhizoma* | 1.00 |
| *Magnoliae Flos* | 1.00 |

calculate the total score as the sum of the 20 questions. At visit 2, the participants will be given a daily diary, including the WURSS-21-K, and instructed to write in it every day and return the diary at visit 3.

**Secondary outcome measures.** The secondary measures are the changes in the total, symptom, and quality of life scores of the WURSS-21-K, nasal congestion severity score, and nasal and systemic symptom score visual analogue scale (VAS) at every time point compared with the baseline. The WURSS-21-K score, nasal congestion severity score, nasal and systemic symptom VAS score, and duration of common cold symptoms are assessed every day through the daily diary. These outcomes will be recorded in the daily diary given to the participants, as described above.

### 1) Total, symptom, and quality of life scores of the WURSS-21-K

The changes in the total score of WURSS-21-K at every time point (except for day 7) compared with the baseline are evaluated. Although the changes in total score of WURSS-21-K at day 7 compared with baseline are assessed as the primary outcome measures, we evaluated the scores at every time point, considering that the common cold is self-limiting disease and its symptoms may change greatly from day to day. Changes in symptoms and quality of life scores of the WURSS-21-K are also evaluated. The symptom score of WURSS-21-K is the sum of scores in response to 10 questions regarding symptoms: runny nose, plugged nose, sneezing, sore throat, scratchy throat, cough, hoarseness, head congestion, chest congestion, and feeling tired. The quality of life score of the WURSS-21-K is calculated as the sum of the scores of nine questions concerning quality of life: think clearly, sleep well, breathe easily, walk/climb stairs/exercise, accomplish daily activities, work outside the home, work inside the home, interact with others, and live personal life.

### 2) Nasal congestion severity score

The nasal congestion severity score assesses the severity of nasal congestion from 0 to 3 score (0 = no symptom; 1 = symptom clearly present but minimal awareness; 2 = definite awareness of symptom, which is bothersome but tolerable; 3 = symptom that is hard to tolerate, may cause interference with activities of daily life or sleeping) [29]. As mentioned earlier, to be included in the clinical trial, an individual must have scored at least 2 at screening. While the WURSS-21-K is a questionnaire used to assess improvements in the overall symptoms of the common cold, the nasal congestion severity score is used as a more specific measure of nasal congestion and the inclusion criteria in this trial.

### 3) Nasal symptom score VAS

In order to evaluate the severity of nasal symptoms in greater detail, a VAS score is used. The nasal symptom score VAS evaluates the severity of nasal symptoms (runny nose, nasal congestion, and sneezing) by marking a VAS labeled number from 0 to 100 (0 = "no symptoms" and 100 = "the worst possible symptoms").

### 4) Systemic symptom score VAS

The systemic symptom score VAS assesses the severity of systemic symptoms (headache, body aches, and chills) by marking a VAS labeled number from 0 to 100 (0 = "no symptoms" and 100 = "the worst possible symptoms"). Because GGT has been used to treat fever, headache, and neck stiffness [21], the GGTCS is expected to be effective in treating systemic symptoms.

### 5) Duration of common cold symptoms

Among the questions included in the daily diary given to participants, there is the question of whether the symptoms of the common cold had disappeared. If the participant answers on any particular day that the symptoms have disappeared, the duration of symptoms will be noted.

### 6) Recovery from common cold

Recovery from the common cold is evaluated based on whether all symptoms of the common cold have disappeared by the time of visit 3.

## 7) Global evaluation of efficacy

The global evaluation of efficacy is a rating scale used to evaluate overall satisfaction with treatment, ranging from 0 to 4 (0 = very good; 1 = good; 2 = moderate; 3 = bad; 4 = very bad), and will be assessed at visit 3 [30].

## 8) 5-level EuroQol 5-dimensional questionnaire (EQ-5D-5L)

The EQ-5D-5L is a self-rating questionnaire evaluating the quality of life in the general population or in patients with various diseases, and consists of five dimensions (mobility, self-care, usual activities, pain/discomfort, and anxiety/depression), with five levels for each question. Also, participants rate their health by marking a vertical VAS labeled number from 0 to 100 (100 means "the best health you can imagine" and 0 means "the worst health you can imagine") [31]. The EQ-5D-5L will be evaluated at visits 2 and 3, and the changes in the EQ-5D-5L score on day 7 will be compared with the baseline.

### Exploratory outcome measures

A questionnaire for common cold pattern identification (QCCPI) will be used to identify patterns of the common cold at visit 1. The QCCPI consists of questions reported by the participants themselves and those completed by the interviewers [32]. The eight patterns of common cold are classified as: Wind-cold pattern, Wind-heat pattern, Contain summer-heat pattern, Contain dampness pattern, Qi deficiency pattern, Blood deficiency pattern, Yin deficiency pattern, and Yang deficiency pattern. In this trial, the patterns of common cold are classified as three patterns: Wind-cold pattern, Wind-heat pattern, and other pattern (any of the other six patterns). The pattern identification is used to determine appropriate treatment for patients' characteristics in Korean medicine. Based on the results of QCCPI, we will investigate the correlation between patterns of common cold and the efficacy of GGTCS.

### Safety assessment

In the safety assessment, adverse events (AEs), vital signs, electrocardiograms, and the results of clinical laboratory tests (complete blood count and liver function test) will be assessed. AEs are harmful and unintended signs, symptoms, and diseases in patients taking investigational drugs, which do not necessarily imply a causal relationship with the investigational drugs. Vital signs and AEs will be assessed at visits 2 and 3. The investigators will record the date of onset, severity, results of AEs, action taken with respect to the investigational drugs, relationship between AEs and investigational drugs, other suspected medicines, and treatments for AEs. When AEs occur, the principal investigator should conduct continuous follow-up until the AEs are resolved or the patient is in a stable condition. All serious AEs should be reported to the principal investigator within 24 hours.

### Allocation

The participants are allocated according to blocked randomization without stratification. A randomized allocation sequence will be assigned to each group based on a computer-generated random number table created by an independent statistician using SAS version 9.4 statistical software (SAS Institute. Inc., Cary, NC, USA). After consenting to the clinical trial, identification codes will be assigned to participants who meet the inclusion and exclusion criteria. The participants will be allocated to the GGTCS or placebo group at an allocation ratio of 1:1. To maintain blinding, the randomization table will be retained separately by the independent statistician. The assignment of participants will be conducted according to the assignment table generated by a random assignment method that can be specifically planned and reproduced in advance. The assigned identification codes will be kept in an opaque envelope and a cabinet with double locks.

### Blinding

As this clinical trial is double-blinded, the allocation of participants will be blinded to both the participants and investigators until the clinical trial is completed. The investigators will not be involved in the randomization and will not know what type

of drugs the participants will take. The GGTCS and placebo will be packed in the same wrapping so that the participants are unaware which medicine they are given. Identification codes will be developed to prevent participants from distinguishing between their assigned groups. Participants will be assigned random numbers and will receive medication packages corresponding to the identification code. The investigators will manage the allocation of identification codes, which should not be disclosed until the clinical trial is completed, except in cases of emergency, when the identification code will need to be disclosed.

## Data management and monitoring

Documents and records related to the clinical trial should be kept in place with locks and stored on a computer with restricted access to those irrelevant to the trial to maintain security. Investigators should maintain essential documents and documents related to the clinical trial, including electronic documents, through relevant acts. After the clinical trial report is completed, these documents should be transferred to a keeping manager, who should preserve the documents related to the clinical trial during the three years following the end of the trial. To ensure quality of the trial, the investigators will be instructed to understand the protocol and manage unexpected AEs, abiding by Korean Good Clinical Practice (KGCP).

Monitoring will be conducted via regular visits and occasional telephone calls. Monitoring includes inspecting compliance with the protocol, collection of appropriate and accurate data, consent of participants, recording and collection of AEs, and management of investigational drugs. When monitors visit, they check the original records of the subjects, medication inventory, dispensing logs, and data storage. Monitoring will be conducted according to the monitoring plan, which includes the strategy, method, responsibility, and monitoring requirements. The monitors will check the progress of the clinical trials, and problems in the trial will be discussed with the investigators.

## Statistical analysis

In this trial, an efficiency analysis will be mainly conducted using a full analysis set (FAS), and a per-protocol set (PPS) analysis will be performed as needed. The FAS group is defined as the analysis group based on intention-to-treat (ITT) principles. Participants in the FAS group will receive the intervention at least once when evaluating efficiency. All data will be included in the analysis after excluding participants who do not meet the following exclusion criteria: those who violate the inclusion/exclusion criteria, have never taken investigational medicines, or do not provide any data after screening. The PPS analysis group consists of participants who complete the entire trial course with ≥70% compliance to investigational medicines without violating the protocol.

All statistical analyses in this trial will be conducted as a two-tailed test, with the significance level set to 5%, using SAS version 9.4 statistical software (SAS Institute. Inc., Cary, NC, USA). The mean and 95% confidence interval for continuous data, and frequency and percentage for categorical data are presented. If necessary, a subgroup analysis will be conducted by categorization of early characteristic at screening or baseline visit. In this trial, missing data will be substituted with Last Observation Carried Forward (LOCF). The substitution of missing data will be applied to the outcome measures in FAS group.

Regarding demographic characteristics and baseline data, the descriptive statistics will be presented for each group. Continuous variables will be expressed as means and confidence intervals, and an independent *t*-test or Wilcoxon rank-sum test will be performed depending on their normality. Categorical variables will be expressed as frequencies and percentages, and Chi-square test or Fisher's exact test will be conducted.

The primary outcome measures will be analyzed using an analysis of covariance (ANCOVA), with treatment group as a fixed effect and the total WURSS-21-K score (symptom score + quality of life score) at baseline as a covariate. If necessary, variables with significant differences in demographic characteristics or those that can affect the symptoms of the common cold can be included as covariates. In addition, the least-square mean (LSM) of the differences between the

GGTCS and control groups is calculated, and the 95% confidence interval (CI) and *p*-values of the differences are presented. Randomization was stratified by center, and potential center effects were considered at the design stage; therefore, center was not included as a covariate in the primary analysis.

Among the secondary outcome measures, analyses were exploratory and interpreted accordingly. The changes in symptom score and quality of life score in the WURSS-21-K, nasal and systemic symptom score VAS, EQ-5D-5L score from baseline to day 7, and duration of the common cold will be analyzed using the same methods as the primary outcome measures. Depending on normality, Student's paired *t*-test will be used to analyze differences in scores before and after treatment. Repeated-measures analysis of variance (RM ANOVA) will be used to compare the differences in the changes in trends between groups, and Dunnett's procedure for multiple comparison correction will be used.

In safety evaluation, the frequencies of AEs and serious AEs suspected of being associated with treatment will be analyzed. AEs are collected through reports of participants or observation of investigators. AEs are recorded in descriptive method with detailed explanation. The frequencies of AEs with relations to interventions of the clinical trial and those without relations are recorded and presented as descriptive statistics. The frequencies of AEs between groups will be compared using Fisher's exact test. The differences in variables in laboratory test before and after treatment are analyzed using paired t-test.

### Ethic and dissemination

This trial follows the principles of the Declaration of Helsinki and has been approved by the Institutional Review Board (IRB) of Daejeon University Daejeon Korean Medicine Hospital (approval no. DJDSKH-24-DR-10), Kyung Hee University Korean Medicine Hospital (approval no. KOMCIRB IRB 2024-10-010), and Pusan University Korean Medicine Hospital (approval no. PNUKHIRB IRB 2024-10-009). Before the clinical trial, the investigators will provide all information related to the trial through written informed consent forms. Participants should be allowed sufficient time to obtain their approval. After the participants voluntarily decide to participate in the trial, they will sign a document containing all the instructions.

This protocol is version 1.3, registered on February 28, 2025 at the National Clinical Trial Registry Clinical Research Information Service (https://cris.nih.go.kr) with identifier number KCT0010251. The results of this trial will be published in a peer-reviewed journal.

### Trial status

The recruitment of participants began in February 2025, and the first participant was enrolled on 28 February, 2025. The recruitment period for this study is expected to be from February 2025 to November 2026. Data collection is anticipated to be completed in January 2027, and the results of this study will be available by April 2027.

### Discussion

The common cold is a viral infection characterized by symptoms related to the upper respiratory tract (runny nose, nasal congestion, sneezing, and cough) and systemic symptoms (headache, body aches, fever, and chills) [1]. Although it is a self-limiting and mild syndrome, it can disrupt daily activities and cause economic losses due to medical expenses and productivity reduction [2,4]. In particular, patients with the common cold find nasal symptoms such as nasal congestion to be the most bothersome symptoms [33]. Medications that alleviate the nasal symptoms of the common cold, including antihistamines and decongestants, have been used. However, they can cause adverse effects such as drowsiness, decreased concentration, nausea, and vomiting, making long-term use difficult [12,13]. The aim of this clinical trial is to investigate the efficacy and safety of GGTCS as an alternative treatment for common cold with nasal congestion.

GGT has been used to treat the common cold, fever, headache, and viral infectious diseases and is composed of seven medicinal herbs: *Puerariae Radix*, *Ephedrae Herba*, *Cinnamomi Ramulus*, *Paeoniae Radix*, *Glycyrrhizae Radix et Rhizoma*, *Zingiberis Rhizoma*, and *Zizyphi Fructus* [21]. Previous studies demonstrated that GGT had antipyretic and analgesic effects

*in vivo* [34,35]. GGT also has an anti-allergic effect, inhibiting the vascular permeability response to serotonin and histamine, and contact dermatitis induced by picryl chloride [35]. GGT reduced expression of proinflammatory cytokines such as IL-1α, IL-6, and TNF-α and improved the Th1/Th2 imbalance related to the inflammation [36]. GGT also had antiviral effects on influenza virus (H1N1), herpes simplex virus 1 (HSV-1), and human respiratory syncytial virus (HRSV) [37–39].

GGTCS is a modified herbal formula of GGT, with *Cnidii Rhizoma* and *Magnoliae Flos* added to GGT. GGTCS has been used to treat allergic rhinitis, sinusitis, and the common cold with nasal congestion and improves inflammation and congestion in the nasal mucosa [22,23]. *Cnidii Rhizoma* has anti-inflammatory and antioxidative effects with an increase in blood flow and is used for headache and pain [40,41]. *Magnoliae Flos* is a medicinal herb used for the treatment of rhinitis, sinusitis, and headache, and its anti-allergic, anti-inflammatory, and antimicrobial activities have been reported [42]. Considering that mucosal inflammation is the primary pathophysiological mechanism of nasal congestion, which leads to increased nasal secretions and tissue swelling [43], GGTCS is expected to possess therapeutic potential for treating nasal congestion and other symptoms of the common cold.

According to the clinical trial guideline for the use of Korean herbal medicine in the treatment of the common cold, the recommended administration period should not exceed eight days [44]. Previous randomized controlled trials using herbal medicine for the common cold have shown that while the significant differences compared with placebo group were observed at day 6–7, these differences diminished after eighth day [16,17]. Therefore, we set the treatment period to seven days to ensure clear assessments and adherence to the guideline. Further, to evaluate efficacy at every time point, we will also analyze daily scores of WURSS-21-K.

While previous studies on GGTCS have primarily targeted allergic rhinitis or chronic rhinosinusitis, there remains a lack of studies focusing specifically on nasal congestion associated with acute viral infections. Consequently, the efficacy of GGTCS for patients with the common cold cannot be readily generalized based on the existing evidence. Although a randomized controlled trial protocol investigating the efficacy and safety of GGTCS for chronic rhinosinusitis has been published, it was limited by a single-center design and a small sample size ($n = 58$) [25]. In contrast, this study is a randomized, double-blind, placebo-controlled, multi-center clinical trial conducted across three sites with a larger sample size ($n = 176$). These methodological strengths are expected to enhance external validity and generalizability, providing more robust and reliable evidence. Furthermore, by assessing daily symptoms severities, including WURSS-21-K score, nasal congestion severity score, nasal and systemic symptom VAS score, and duration of common cold, this study enables detailed analysis of symptom changes over time. In addition, the assessment of the treatment satisfaction and the common cold patterns offers multidimensional perspective on treatment effects. Satisfaction outcomes may provide insights into treatment adherence to GGTCS. Moreover, if association between the common cold patterns and the efficacy of GGTCS are identified, the results would highlight the potential for a precision medicine within Korean medicine, facilitating more effective and individualized treatment for patients with the common cold.

However, there are some inherent limitations with this study. First, diagnostic tests to confirm other respiratory infectious diseases such as influenza and COVID-19 will not be conducted for this trial. Although participants over 38°C will be excluded from our study, and participants' symptoms, vital signs, radiological findings, and results of laboratory test will be checked to rule out other infectious diseases, it is possible to include participants with other respiratory infectious diseases with mild symptoms in this trial. However, we followed the guide of interstitial lung disease and symptoms similar to COVID-19 to exclude those diseases, as also done in real clinical practice. Second, all of the outcome measures are obtained from self-report questionnaires by participants without objective assessments, as clinical improvements of the common cold are usually measured by symptoms of each patient. However, it still has limitation for inducing self-report bias in evaluating their severity and duration of symptoms. After conducting this trial, we will evaluate the feasibility to use objective assessments as nasal endoscopy or nasal airway resistance. Lastly, we will not include children in this trial, despite high prevalence of the common cold in children. After analyzing the results of this trial, we will conduct further clinical trials targeting children.

In previous clinical trials in which the efficacy and safety of herbal medicines has been assessed, the focus has been the overall symptoms of the common cold [16,17,45]; we are not aware of any clinical trials with a focus on specific symptoms of the common cold, in particular nasal symptoms. The common cold causes various local and systemic symptoms, and its prognosis and treatment of the common cold vary according to the type of symptoms. As conventional medicine for the common cold is prescribed in accordance with patients' symptoms [10], it is necessary to conduct studies on herbal medicine with subdivisions according to the symptoms of the common cold. Nasal congestion is one of the common symptoms of the common cold and distresses patients by inducing discomfort in the nose, headache, sleep disturbance, fatigue, and poor concentration, which may decrease the quality of life and productivity [19,20]. Therefore, developing treatments to treat nasal congestion caused by the common cold is clinically meaningful. This study aimed to investigate the effectiveness of GGTCS for nasal congestion with the common cold. If confirmed, the results of this trial could provide a basis for the design of further studies and the application of herbal medicine in nasal symptoms accompanied by respiratory infections.

This study was designed to demonstrate the effects and safety of GGTCS on the symptoms of nasal congestion associated with the common cold. Considering that GGTCS has been used to treat upper respiratory diseases with nasal symptoms and that previous studies have reported its therapeutic effects and mechanisms, GGTCS is expected to have clinical potential to alleviate the symptoms of the common cold, especially nasal congestion. Moreover, we will evaluate the pattern identification used for diagnosis in Korean medicine to analyze its correlation with effectiveness. The results of this clinical trial are expected to provide evidence for the effectiveness and safety of GGTCS for nasal congestion associated with the common cold and help facilitate the clinical use of herbal medicine in patients with the common cold.

## Conclusion

In conclusion, this clinical trial will provide evidence for the effectiveness and safety of GGTCS for nasal congestion associated with the common cold and help facilitate the clinical use of herbal medicine in patients with the common cold.

## Supporting information

**S1 File. SPIRIT checklist.**
(DOCX)

**S2 File. Study protocol (English).**
(PDF)

## Acknowledgments

We thank all the research staff at Daejeon University Korean Medicine Hospital, Kyung Hee University Korean Medicine Hospital, and Pusan University Korean Medicine Hospital for their contributions to this trial.

## Author contributions

**Conceptualization:** Yee Ran Lyu, Yang-Chun Park.

**Formal analysis:** Byong-Kab Kang.

**Investigation:** Bok-Nam Seo, Ae-Ran Kim, Boram Lee.

**Supervision:** Yang-Chun Park.

**Writing – original draft:** Seong-Cheon Woo.

**Writing – review & editing:** Yee Ran Lyu, Byong-Kab Kang, Bok-Nam Seo, Ae-Ran Kim, Boram Lee, Yang-Chun Park.

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
