## [Decision Letter · Decision Letter 0]

21 Dec 2025

Dear Dr. Park,

Thank you for submitting your manuscript to PLOS ONE. After careful consideration, we feel that it has merit but does not fully meet PLOS ONE’s publication criteria as it currently stands. Therefore, we invite you to submit a revised version of the manuscript that addresses the points raised during the review process.

We look forward to receiving your revised manuscript.

Kind regards,

Rajeev Singh

Academic Editor

PLOS One

Journal Requirements:

This research was supported by a grant from the Korea Health Technology R&D Project through the Korea Health Industry Development Institute (KHIDI), funded by the Ministry of Health & Welfare, Republic of Korea (RS-2024-00441852).

Reviewers' comments:

Reviewer's Responses to Questions

**Comments to the Author**

1. Does the manuscript provide a valid rationale for the proposed study, with clearly identified and justified research questions?

Reviewer #1: Yes

Reviewer #2: Partly

2. Is the protocol technically sound and planned in a manner that will lead to a meaningful outcome and allow testing the stated hypotheses?

Reviewer #1: Yes

Reviewer #2: Partly

3. Is the methodology feasible and described in sufficient detail to allow the work to be replicable?

Reviewer #1: Yes

Reviewer #2: Yes

4. Have the authors described where all data underlying the findings will be made available when the study is complete?

Reviewer #1: Yes

Reviewer #2: No

5. Is the manuscript presented in an intelligible fashion and written in standard English?

Reviewer #1: Yes

Reviewer #2: Yes

You may also provide optional suggestions and comments to authors that they might find helpful in planning their study.

Reviewer #1: As the statistical reviewer I will focus on methods and reporting.

Major

1) The power calculation does not seem to have accounted for centre heterogeneity. is there no ICC expected? Also it is impossible to quantify what the assumed mean difference and SD mean, without a description of the disstribution of the outcome. is 30 small? medium? large effect? So the authors probably need to quantify using Cohen's d or a similar measure.

2) the analysis plan is a t-test, which implies follow-up comparison. Anyway, the plan is broadly acceptable and the authors plan to use ANCOVA if there is imbalance. However, I would advise mixed effects modelling (linear or logistic regression), with a random intercept for centre (to account for heterogeneity): with covariates if there is imbalance. this will allow to control for covariates if there is imbalance while moving away from p-values alone and focus on effect sizes (MDs) and their associated confidence intervals.

3) I disagree with the LOCF approach which has been shown to be biased. using a multiple imputation approach is preferable and should be a sensitivity analysis as a minimum.

4) analyses for secondary outcomes need to be treated as exploratory: power may be limited and there are multiple comparisons, inflating type-I error.

Minor

1) The authors in the power calculations section say "the changes in total WURSS-21-K (Wisconsin Upper Respiratory Symptom Survey-21-Korean version) score measured at day 7 compared to the baseline between GGTCS and placebo group". How will the comparison to baseline take place? in RCTs the assumption is that baseline is the same in the two arms, and the outcome is compared between 2 groups in the follow-up time points.

2) t-tests assume normality and the authors have not discussed that - however see my major point 2, moving to regression models resolves this as well.

Reviewer #2: 1. Need for general background on conventional treatments for nasal congestion

In the early part of the Introduction, the manuscript discusses general aspects of the common cold. However, before introducing herbal interventions (line ~78), it would be beneficial to include standard conventional treatments for nasal congestion—such as decongestants, antihistamines, intranasal corticosteroids, and saline rinses—and discuss their limitations.

This will create a more balanced clinical context and reinforce the rationale for exploring GGTCS as an alternative.

2. Clarification needed for the study rationale despite existing evidence of GGTCS efficacy

The manuscript states that GGTCS is already used clinically to alleviate nasal congestion, and that previous studies have shown its immediate effect on nasal obstruction, particularly in allergic conditions.

Given this, it is unclear why this randomized clinical trial is still necessary.

Please clarify:

what specific evidence gap remains,

whether prior evidence is limited to allergic rhinitis rather than viral common cold,

and how this study will provide novel clinical value.

A strengthened rationale will improve the scientific justification of the protocol.

3. Justification for selecting a 7-day intervention period

The administration period for the investigational product is set to 7 days. However, most common cold symptoms—including nasal congestion—naturally improve within this timeframe.

Please provide justification for:

why 7 days was chosen as the treatment duration,

whether previous studies support this specific duration for GGT/GGTCS,

and how the study intends to differentiate the effect of GGTCS from the natural course of recovery.

This clarification is essential to avoid concerns regarding spontaneous symptom resolution.

4. Potential imbalance in participant recruitment across sites

Although the trial is multicenter and double-blinded, the manuscript does not specify whether participant enrollment was balanced across the three hospitals.

Site imbalance can influence treatment effects and generalizability.

Please clarify:

whether site-specific recruitment was monitored,

whether any imbalance is anticipated or occurred,

and whether statistical adjustment for site will be applied if necessary.

5. Absence of results but presence of result-like statements

As a study protocol, the manuscript should not imply or suggest that results have already been obtained.

However, certain parts of the Discussion appear to describe GGTCS as if its efficacy has been confirmed.

Please remove or revise statements that could be misconstrued as results from this trial.

All statements should clearly reference anticipated findings or rationale, not outcomes.

6. Discussion does not sufficiently articulate the potential significance of the study

The Discussion section mainly reiterates that GGTCS is expected to be effective for nasal congestion. However, it lacks a deeper explanation of:

how the results (once available) will provide meaningful clinical insights,

how this trial differs from or improves on previous studies,

and what contributions this study will make to the field of respiratory or herbal medicine.

Strengthening the Discussion to reflect the future implications and unique value of the trial is recommended.

**Do you want your identity to be public for this peer review?** For information about this choice, including consent withdrawal, please see our Privacy Policy

Reviewer #1: No

Reviewer #2: No

---

## [Author Response · Author response to Decision Letter 1]

2 Jan 2026

Journal Requirements:

: We have revised the manuscript according to the style requirements.

: We have revised the ethic statement as “Ethic and dissemination” within the Methods section.

: We have provided Data Availability Statement in the submission form, and revised it in the manuscript as “No data were generated or analyzed during the current study. All relevant data will be made available upon completion of the study.”

This research was supported by a grant from the Korea Health Technology R&D Project through the Korea Health Industry Development Institute (KHIDI), funded by the Ministry of Health & Welfare, Republic of Korea (RS-2024-00441852).

: The funders had no role in study design, data collection and analysis, decision to publish, or preparation of the manuscript.

: The reviewers did not request any specific published works during revision.

Response to Reviewers comments:

Reviewer #1: As the statistical reviewer I will focus on methods and reporting.

Major

1) The power calculation does not seem to have accounted for centre heterogeneity. is there no ICC expected? Also it is impossible to quantify what the assumed mean difference and SD mean, without a description of the disstribution of the outcome. is 30 small? medium? large effect? So the authors probably need to quantify using Cohen's d or a similar measure.

: We thank the reviewer for the helpful comments regarding the sample size calculation, effect size assumptions, and centre heterogeneity.

First, regarding the magnitude of the assumed treatment effect, we agree that reporting only the absolute mean difference and standard deviation may limit interpretability without standardization. Based on the assumed mean difference of 10 points and a standard deviation of 21 points, the corresponding standardized effect size (Cohen’s d) is approximately 0.48, which represents a moderate effect size according to conventional criteria. This assumption was informed by prior randomized controlled trials using the WRUSS-21-K score and reflects a clinically meaningful treatment effect. The sample size calculation was conducted using the assumed mean difference and standard deviation on the original WRUSS-21-K scale, while Cohen’s d is provided to facilitate interpretation of the magnitude of the assumed effect.

We also note that a value of 30 points was inadvertently stated in an earlier version of the manuscript when referring to the mean difference. We apologize for this clerical error and have corrected it accordingly. The correct assumed mean difference used for the sample size calculation was 10 points, and this correction does not affect the sample size estimation or the study conclusions. (Line 179)

Regarding centre heterogeneity, randomization was performed at the individual level with stratification by centre to account for potential centre-level heterogeneity at the design stage and to ensure balance between treatment groups within each centre. As the study was not cluster-randomized and all centres followed an identical protocol, substantial intra-centre correlation was not anticipated; therefore, an intraclass correlation coefficient (ICC) was not included in the sample size calculation.

Overall, the sample size determination was guided by prior evidence and focused on estimating a clinically meaningful treatment effect, with centre effects addressed through the study design.

2) the analysis plan is a t-test, which implies follow-up comparison. Anyway, the plan is broadly acceptable and the authors plan to use ANCOVA if there is imbalance. However, I would advise mixed effects modelling (linear or logistic regression), with a random intercept for centre (to account for heterogeneity): with covariates if there is imbalance. this will allow to control for covariates if there is imbalance while moving away from p-values alone and focus on effect sizes (MDs) and their associated confidence intervals.

: We thank the reviewer for the helpful comments regarding the statistical analysis plan.

We would like to clarify that the primary efficacy analysis was prespecified as an analysis of covariance (ANCOVA) with baseline adjustment, whereas independent t-tests or Wilcoxon rank-sum tests were used only for descriptive comparisons of baseline characteristics.

We appreciate the reviewer’s suggestion regarding the use of mixed-effects modelling. In the present study, randomization was performed at the individual level with stratification by centre to control for potential centre-level heterogeneity at the design stage and to ensure balance between treatment groups within each centre. Accordingly, centre effects were addressed through the study design and were not included as covariates in the primary analysis. All participating centres followed an identical protocol with the same intervention and outcome assessment.

As specified in the manuscript, relevant covariates may be included in the ANCOVA model if baseline imbalances are identified, and treatment effects will be reported as adjusted mean differences with corresponding 95% confidence intervals. (Line 389-390)

3) I disagree with the LOCF approach which has been shown to be biased. using a multiple imputation approach is preferable and should be a sensitivity analysis as a minimum.

: We thank the reviewer for the thoughtful comment regarding the handling of missing data and the use of the last observation carried forward (LOCF) approach. We acknowledge the reviewer’s concern regarding the potential bias of LOCF. Given that the study included only two study visits (baseline and day 7), the LOCF approach in this context is equivalent to a baseline observation carried forward approach and therefore represents a conservative assumption that may underestimate treatment effects rather than inflate them. Nevertheless, to further assess the robustness of the findings, a multiple imputation approach may be explored as a sensitivity analysis.

4) analyses for secondary outcomes need to be treated as exploratory: power may be limited and there are multiple comparisons, inflating type-I error.

: We thank the reviewer for the comment regarding the interpretation of secondary outcome analyses.

We agree that analyses of secondary outcomes should be interpreted as exploratory, as the study was primarily powered for the primary endpoint. Accordingly, secondary outcome results are presented as supportive and exploratory findings.

We would also like to clarify that, as specified in the Statistical Analysis section, Dunnett’s procedure for multiple comparison correction is applied when comparing secondary outcomes with the control group. This prespecified approach controls the family-wise type-I error rate arising from multiple comparisons and therefore addresses concerns regarding inflation of type-I error. (Line 391)

Minor

1) The authors in the power calculations section say "the changes in total WURSS-21-K (Wisconsin Upper Respiratory Symptom Survey-21-Korean version) score measured at day 7 compared to the baseline between GGTCS and placebo group". How will the comparison to baseline take place? in RCTs the assumption is that baseline is the same in the two arms, and the outcome is compared between 2 groups in the follow-up time points.

: We thank the reviewer for pointing out the potential ambiguity in the wording of the primary outcome description in the power calculation section.

The original phrasing may have been misinterpreted as implying a direct comparison of baseline values between treatment groups. This was not the intended meaning. The primary outcome of interest is the between-group difference in the total WURSS-21-K score measured at day 7, with baseline values used solely for adjustment.

To avoid misunderstanding, we have revised the wording of the primary outcome description in the manuscript to clarify that the primary outcome is the between-group difference in the total WURSS-21-K score measured at day 7, adjusted for baseline values. (Line 174-176)

2) t-tests assume normality and the authors have not discussed that - however see my major point 2, moving to regression models resolves this as well.

: We thank the reviewer for the comment regarding the normality assumption underlying t-tests.

We would like to clarify that independent t-tests were used only for descriptive comparisons, whereas the primary efficacy analysis is based on a model-based approach (ANCOVA). As such, the primary analysis does not rely on a simple two-sample t-test at follow-up.

In addition, the ANCOVA framework can be viewed as a linear regression model, which is generally robust to moderate departures from normality, particularly with the sample sizes planned in this study. Therefore, concerns regarding strict normality assumptions are mitigated in the primary analysis.

Accordingly, while normality assumptions are considered in descriptive analyses, the primary treatment effect estimation is conducted using a regression-based approach, as also noted in the manuscript.

Reviewer #2:

1. Need for general background on conventional treatments for nasal congestion

In the early part of the Introduction, the manuscript discusses general aspects of the common cold. However, before introducing herbal interventions (line ~78), it would be beneficial to include standard conventional treatments for nasal congestion—such as decongestants, antihistamines, intranasal corticosteroids, and saline rinses—and discuss their limitations.

This will create a more balanced clinical context and reinforce the rationale for exploring GGTCS as an alternative.

: We thank the reviewer for the comment regarding background of conventional treatment for nasal congestion.

We have added the limitations of conventional treatments (intranasal corticosteroids and nasal irrigation). We have highlighted their potential side effects in the Introduction. (Line 70-73)

2. Clarification needed for the study rationale despite existing evidence of GGTCS efficacy

The manuscript states that GGTCS is already used clinically to alleviate nasal congestion, and that previous studies have shown its immediate effect on nasal obstruction, particularly in allergic conditions.

Given this, it is unclear why this randomized clinical trial is still necessary.

Please clarify:

what specific evidence gap remains,

whether prior evidence is limited to allergic rhinitis rather than viral common cold,

and how this study will provide novel clinical value.

A strengthened rationale will improve the scientific justification of the protocol.

: We thank the reviewer for pointing out the study rationale and evidence gap.

We have revised the Introduction to clarify the distinct clinical value of this study.

Because previous studies on GGTCS targeted allergic rhinitis or chronic rhinosinusitis. However, the pathophysiology of nasal congestion in the common cold differs from allergic condition. Therefore, evidence from these previous studies cannot be directly applied to the common cold.

The previous studies are limited to case reports and a protocol for randomized controlled trial (RCT) for chronic rhinosinusitis (the result of the RCT is not yet published). There is a lack of high-quality, RCTs providing robust clinical data.

Considering these previous studies, the present study is expected to bridge the evidence gap, targeting nasal congestion associated with acute viral infections (common cold) through RCT. (Line 92-98)

3. Justification for selecting a 7-day intervention period

The administration period for the investigational product is set to 7 days. However, most common cold symptoms—including nasal congestion—naturally improve within this timeframe.

Please provide justification for:

why 7 days was chosen as the treatment duration,

whether previous studies support this specific duration for GGT/GGTCS,

and how the study intends to differentiate the effect of GGTCS from the natural course of recovery.

This clarification is essential to avoid concerns regarding spontaneous symptom resolution.

: We thank the reviewer for pointing out the justification of intervention period.

The 7-day duration was determined based on the 'Clinical Trial Guidelines for Korean Herbal Medicine in Common Cold,' which recommends an administration period not exceeding eight days. Two previous RCTs using herbal medicine for the common cold have shown that while the significant differences compared with placebo group were observed at Day 6 to Day 7, these differences were no longer significant after Day 8.

Based on the references, we set the treatment period to seven days, and we expected that the 7-day treatment would ensure clear assessments and adherence to the guideline.

Furthermore, to distinguish the effect of GGTCS from spontaneous resolution, we have employed daily monitoring using the WURSS-21-K. By analyzing the daily symptom scores, we aim to demonstrate that GGTCS leads to a faster and more significant reduction in symptoms compared to the natural course observed in the placebo group. (Line 454-460)

4. Potential imbalance in participant recruitment across sites

Although the trial is multicenter and double-blinded, the manuscript does not specify whether participant enrollment was balanced across the three hospitals.

Site imbalance can influence treatment effects and generalizability.

Please clarify:

whether site-specific recruitment was monitored,

whether any imbalance is anticipated or occurred,

and whether statistical adjustment for site will be applied if necessary.

: We thank the reviewer for the comment regarding the potential imbalance in participant recruitment.

In this study, randomization was performed at the individual level with stratification by centre to account for potential centre-level heterogeneity at the design stage and to ensure balance between treatment groups within each centre. To minimize inter-center variability, we conducted standardized training for all investigators based on Standard Operating Procedure (SOP). Regular clinical monitoring is being performed to ensure strict adherence to the protocol across all three sites. (Line 389-390)

5. Absence of results but presence of result-like statements

As a study protocol, the manuscript should not imply or suggest that results have already been obtained.

However, certain parts of the

---

## [Decision Letter · Decision Letter 1]

22 Jan 2026

Efficacy and safety of Galgeun-tang-ga-cheongung-sinyi for nasal congestion with common cold: a study protocol for randomized, double-blind, placebo-controlled, parallel, multicenter clinical trial

PONE-D-25-52070R1

Dear Dr. Park,

We’re pleased to inform you that your manuscript has been judged scientifically suitable for publication and will be formally accepted for publication once it meets all outstanding technical requirements.

Kind regards,

Rajeev Singh

Academic Editor

PLOS One

Additional Editor Comments (optional):

Reviewers' comments:

Reviewer's Responses to Questions

**Comments to the Author**

1. Does the manuscript provide a valid rationale for the proposed study, with clearly identified and justified research questions?

Reviewer #1: Yes

2. Is the protocol technically sound and planned in a manner that will lead to a meaningful outcome and allow testing the stated hypotheses?

Reviewer #1: Yes

3. Is the methodology feasible and described in sufficient detail to allow the work to be replicable?

Reviewer #1: Yes

4. Have the authors described where all data underlying the findings will be made available when the study is complete?

Reviewer #1: Yes

5. Is the manuscript presented in an intelligible fashion and written in standard English?

Reviewer #1: Yes

You may also provide optional suggestions and comments to authors that they might find helpful in planning their study.

Reviewer #1: I am satisfied with the authors' responses and the resulting changes to the paper. I have no further comments to make.

**Do you want your identity to be public for this peer review?** For information about this choice, including consent withdrawal, please see our Privacy Policy

Reviewer #1: No

---

## [Editor Report · Acceptance letter]

PONE-D-25-52070R1

PLOS One

Dear Dr. Park,

I'm pleased to inform you that your manuscript has been deemed suitable for publication in PLOS One. Congratulations! Your manuscript is now being handed over to our production team.

Kind regards,

on behalf of

Dr. Rajeev Singh

Academic Editor

PLOS One